# Polyphenols as Potent Epigenetics Agents for Cancer

**DOI:** 10.3390/ijms231911712

**Published:** 2022-10-03

**Authors:** Peramaiyan Rajendran, Salaheldin Abdelraouf Abdelsalam, Kaviyarasi Renu, Vishnupriya Veeraraghavan, Rebai Ben Ammar, Emad A. Ahmed

**Affiliations:** 1Department of Biological Sciences, College of Science, King Faisal University, Al-Ahsa 31982, Saudi Arabia; 2Centre of Molecular Medicine and Diagnostics (COMManD), Department of Biochemistry, Saveetha Dental College & Hospitals, Saveetha Institute of Medical and Technical Sciences, Saveetha University, Chennai 600 077, Tamil Nadu, India; 3Department of Zoology, Faculty of Science, Assiut University, Assiut 71515, Egypt; 4Laboratory of Aromatic and Medicinal Plants, Center of Biotechnology of Borj-Cedria, Technopole of Borj-Cedria P.O. Box 901, Hammam-Lif 2050, Tunisia; 5Laboratory of Molecular Physiology, Zoology Department, Faculty of Science, Assiut University, Assiut 71515, Egypt

**Keywords:** cancer, polyphenol, epigenetics

## Abstract

Human diseases such as cancer can be caused by aberrant epigenetic regulation. Polyphenols play a major role in mammalian epigenome regulation through mechanisms and proteins that remodel chromatin. In fruits, seeds, and vegetables, as well as food supplements, polyphenols are found. Compounds such as these ones are powerful anticancer agents and antioxidants. Gallic acid, kaempferol, curcumin, quercetin, and resveratrol, among others, have potent anti-tumor effects by helping reverse epigenetic changes associated with oncogene activation and tumor suppressor gene inactivation. The role dietary polyphenols plays in restoring epigenetic alterations in cancer cells with a particular focus on DNA methylation and histone modifications was summarized. We also discussed how these natural compounds modulate gene expression at the epigenetic level and described their molecular targets in cancer. It highlights the potential of polyphenols as an alternative therapeutic approach in cancer since they modulate epigenetic activity.

## 1. Introduction

Epigenetics is the study of how genes and their products affect an organism’s phenotype [1]. Since the discovery of DNA, epigenetics has not received much attention. In the 1980s, however, studies of chromatin structure made epigenetics respectable after being shrouded in the shadows [2]. In 1987, Robin Holliday redefined epigenetics as nuclear inheritance without DNA sequence differences [3]. It offers possible explanations for cellular differentiation and parental imprinting in mammals, and it enables genetics and developmental embryology to be integrated [4]. Furthermore, epigenetic modifications play a crucial role in developmental patterning, biological processes, and pathology [5,6,7]. In mammalian cells, DNA methylation and histone modifications induce chromatin remodeling, leading to cellular phenotype changes [8,9,10,11]. Diverse epigenetic changes occur in cancer cells in the early stages of tumor development [12]. These epigenetic modifications of chromatin are inherited and reversible, so they could be used to develop drugs targeting the epigenome which could help treat cancer [13,14,15]. The use of new therapeutic drugs and personalized treatment leads to improved patient survival. Dietary supplements have been combined with some of these treatments [16,17,18]. Diets high in vegetables and fruits have been proven to reduce the risk of cancer. This is because they regulate the expression of oncogenes and tumor suppressor genes [19]. Dietary supplements might be an alternative cancer treatment. In this study, we provide an overview of the most common epigenetic alterations seen in cancer, then discuss the most studied dietary patterns and how they might be used to reverse epigenetic changes, and lastly, discuss how they might be used as an alternative therapy for cancer.

## 2. Oxidative DNA Damage and Polyphenols

Polyphenols have antioxidant properties which as a therapeutic action against cancer. It is found that polyphenol has a dominant antioxidant that mitigates oxidative stress from pathological conditions such as cancer. Polyphenol can scavenge ROS and act on free radicals. This is due to the presence of aromatic rings, the presence of hydroxyl groups in a different region, and has more conjugated system [20]. Polyphenol scavenges Reactive Oxygen Species (ROS) and mitigates the biomolecule oxidative damage [21,22]. Polyphenols with antioxidant capacity suppress the signaling pathways involved in oxidative stress generation at a molecular level. Consumption of a polyphenol diet increases the activity of the antioxidant and inhibits the peroxidation of lipids and cyclooxygenase (COX) pathways [23]. Increased levels of free radical production such as ROS and LPO with oxidative stress cause damage to the tissues inclusive of DNA and increases the possibility of cancer occurrence. Increased ROS level is due to exogenous, antioxidant defense, and endogenous sources. The exogenous sources include X-rays, UV light irradiation, the action of metals, toxins, and γ rays, drugs, and solvents; endogenous sources include peroxisomes, metabolism of cytochrome P450; reactions in mitochondria; and activation of inflammation. This exogenous and endogenous source of ROS is important for oxidative stress-mediated ROS production causes damage to the cell, and alters the signaling pathway, which further causes cancer [24]. Damages to the DNA can cause errors in the replication, arresting transcriptional activities, instability of the DNA damage, and further causes cancer [25,26]. Different studies show a reduction in endogenous DNA damage and protection from ex vivo DNA damage [27]. A diet such as vegetables and fruits has a high content of polyphenols which includes quercetin, ellagic acid, catechins, naringenin, and resveratrol. This polyphenol can decrease the risk of cancer. Polyphenols have a chemopreventive action which includes the involvement of antiestrogenic, arresting cell cycle, proliferation against cancer cells, resistance to the oxidative stress, induction of apoptosis, detoxification enzyme activation, regulation of the host immune system, and cellular signaling improvement in the cancer condition [28]. Polyphenols show protection from cancer when combined with DNA-damaging agents. Polyphenols impair the metabolism of pro-carcinogen by altering the level of enzyme cytochrome P450 which plays an important role in the stimulation of cancer [29]. Polyphenol and quercetin have properties of anti-cancer action by reducing free radical-ROS scavenging activity [30]. The polyphenols present in black tea such as theaflavins, EGCG, and thearubigins have effective properties of anti-cancer [31,32,33]. The catechins present in the tea can prevent cancer by impairing intraepithelial prostate lesions converting them into cancer and decreasing the cancer cell apoptosis, thereby it inhibits carcinogenesis [34]. The flavonoids like catechins, anthocyanins, flavanols, flavanones, flavones, and isoflavones, have a capacity for free radicals neutralization via scavenging ROS and impairs the risk of cancer by cellular growth arrest in cancer cells [35]. There are different types of cancer such as prostate, endometrial, epithelial, breast cancer and colon cancer are mitigated by polyphenols [36]. Resveratrol has an anti-cancer property via an antioxidant defense mechanism which impairs the hydroperoxidase level, matrix metalloproteinase level (MMP-9), Akt signaling pathway, NF-KB pathway, cycloxygenase pathway, protein kinase C, Bcl-2 level and focal adhesion kinase [37].

## 3. Human Cancer and DNA Methylation

In cancer, epigenetic changes include genome-scale methylation changes, hypermethylation in specific loci, and dysfunction of histone-modifying enzymes. Changes in DNA methylation are good biomarkers since they can be detected and quantified [23,38,39]. Many studies have found DNA methylation patterns specific to liver cancers, including genome-wide studies [40,41,42,43,44,45]. According to DNA methylation and transcriptome mapping in human tumors, a lot of genes are hypomethylated and expressed more, and a lot of genes are hypermethylated and underexpressed. The genes induced by epigenetics were found to be involved in cellular transformation and differentiation, tumor growth, and metastasis. Apoptosis, cell adhesion, and cell cycle progression genes are repressed [46,47,48]. Even though genome-wide DNA methylation studies are a hot topic, there are a few caveats that urge caution about the clinical and biological significance of the data [49]. Most importantly, tumors have a lot of cellular heterogeneity, so observed differences in DNA methylation patterns might just be due to differences in tumor cell numbers, rather than being an epigenetic signature. The DNA methylation profiling needs to be performed on small numbers of histologically verified tumor cells sorted by high-speed cell sorting or laser dissection microscopy. Another caveat is that we shouldn’t assume a simple relationship between DNA methylation and gene expression, even if transcriptome data indicates that. In vivo experiments would need to manipulate DNA methylation site-directedly and demonstrate transcription rate changes (Table 1).

## 4. Cancer and Histones

DNA is packed into chromatin around an octamer of histones in a chromosome. A nucleosome is a repeating unit of chromatin that is made up of 150 base pairs of DNA and an octamer of histones, H2A, H2B, H3, and H4 [49,50,51,52]. Histone tails are targets for post-translational modifications, including acetylation, methylation, phosphorylation, and ubiquitination [53,54]. DNA modifications can turn the transcription of genes on or off, which affects the accessibility of transcription factors by adjusting how tightly DNA is bound to histones. HATs, which “write” the acetyl mark on histones, are responsible for histone acetylation. By counteracting the positive charge of histones, it loosens the connection between histones and DNA. In contrast, histone deacetylases (HDACs) “erase” those acetyl groups, resulting in tight DNA coiling around the histones, making chromatin transcriptionally inactive. Histone methylation is associated with either transcriptionally active or closed chromatin depending on where the lysine is methylated [55]. The trimethylation of histone 3 lysine 27 (H3K27me3), for example, is associated with transcriptional repression, whereas trimethylation of histone 3 lysine 4 is associated with gene activation [56,57]. Cancer patients with high levels of trimethylated histone H3 lysine 4 (H3K4me3) have a poor prognosis. H3K27me3 levels were linked to poor prognosis and tumor aggressive features including vascular invasion, large tumor sizes, multiple tumors, and poor differentiation in another study [56,58,59]. To fully understand the role of these specific DNA-protein modifications in cancer, further studies using more precise detection methods, such as ChIP-sequencing, will be needed.

## 5. Inhibitors of DNA Methylation

The epigenome is reprogrammed as soon as embryogenesis begins because DNA methylation decreases. Methylation of DNA requires methylating enzymes, so cellular replication without these enzymes leads to significant demethylation of daughter cells and gene reactivation. This approach has a therapeutic ratio when applied to cancer cells; normal cells usually survive hypomethylation, whereas cancer cells usually die when it occurs, perhaps because they are dependent on gene silencing.

DNA hypomethylation only happens with cytosine analogs with 5’ modifications of the ring. Nucleoside and cytosine analogs do not directly affect DNA methylation. It was determined that the ability of these two main analogs to target DNA methyltransferases (DNMTs) for degradation was attributed to their ability to trap DNA methyltransferases (Table 1). In the absence of these enzymes, DNA synthesis results in hypomethylation in daughter cells, which in turn leads to the reactivation of silenced genes. Some other 5’ modified nucleoside analogs have been described in preclinical or early clinical studies [60]. Inhibiting DNA methylation in cancers works, at least in part, by inhibiting DNA methylation.

## 6. Inhibitors of Histone Modification

Inhibitors of histone deacetylase (HDACi) decrease HDAC activity, block acetylated histone aggregation, and promote autosomal acetylation. HDACi promotes cancer cell differentiation, induces apoptosis, and inhibits angiogenesis through many mechanisms, including cell cycle arrest, apoptosis, autophagy, and differentiation. HDACi are classified into four classes based on their chemical. In addition to in vitro and in vivo studies, hydroxamates and aliphatic acids have also been tested in clinical trials as a new treatment strategy for hepatobiliary cancer [61,62,63,64]. Panobinostat, trichostatin A, vorinostat, and belinostat are hydroxamates that block HDAC activity by binding to Zn^2+^ at the HDAC binding site. Besides aliphatic acids, sodium butyrate and valproic acid (VPA) inhibit class I HDACs as well [65,66]. For example, this occurs with VPA, sodium butyrate, and TSA. By downregulating cyclins A and D1 and upregulating P21, VPA could induce cell cycle arrest [67]. Additionally, sodium butyrate upregulated p21 and p27 protein expression [68,69,70]. Furthermore, TSA causes G2/M-phase arrest and G0/G1 arrest in hepatoma cells [71,72]. The fact that apoptosis plays an important role in tumor development makes it an obvious target for cancer therapy. HDACi promotes apoptosis in cancer cells. In addition, HDACi promotes apoptosis by different mechanisms; for example, VPA activates TRAIL-associated cell death and intrinsic apoptosis by upregulating cleaved caspases 3 and 9 [73]. Specifically, TSA upregulates bax and cleaved caspase 3 and downregulates BCL-2 in cancer cells [74,75]. Further, it has been found that HDACi can induce autophagy-mediated cell death, and cancer cell lines showed autophagosome formation, maturation, and aggregation when exposed to panobinostat. Several inhibitors of angiogenesis have been found to interact with HDACi in a synergistic way to inhibit hepatobiliary cancers [76,77]. Some clinical trials have tested HDACi’s anticancer effects, especially when combined with sorafenib.

## 7. Polyphenol and Cancer

There is usually at least one hydroxyl group attached to an aromatic ring in these compounds. Polyphenolic molecules have been found in thousands of higher plants and hundreds of edible plants [78,79]. There are two types of polyphenols in plants: flavonoids and non-flavonoids [80]. Flavanoids share 15 carbon atoms at their core and are divided into flavanols, flavonols, anthocyanidins, flavones, flavanones, and chalcones [81]. The non-flavonoids contain an aromatic ring with one or more hydroxyl groups. Stilbene, phenolic acids, saponin, and other polyphenols such as curcumin and tannins are not flavonoids. Plants synthesize polyphenols to defend themselves against infection and protect themselves from stress. In recent years, plant polyphenols have received substantial attention for their cancer-preventing effects. Several studies have demonstrated that plant polyphenols are chemopreventive against multiple types of cancer [82,83,84,85,86,87].

### 7.1. Epigenetics Mechanism of Kaempferol

The natural flavonoid kaempferol (KFL) is widely found in fruit and veggies. Many cancers, such as breast cancer, pancreatic cancer, prostate cancer, and lung cancer, have been reported as being susceptible to KFL [88,89,90,91]. A previous study has shown that KFL could affect DNA methylation in nude mice treated with bladder cancer [92]. In their experiments, they found that KFL inhibited the levels of DNMT3B protein without affecting DNMT1 and also found that cycloheximide inhibited protein synthesis of DNMT3B and caused DNMT3B to prematurely degrade and block proteasome with MG132, causing KFL to ubiquitinate DNMT3B. Results suggest that KFL can degrade DNMT3B through the ubiquitin-proteasome pathway [93]. Another study showed treatment with KFL increased DACT2 in three colorectal cancer cells. In addition, KFL reduced DACT2 methylation but increased unmethylated DACT2 by binding directly to DNA methyltransferases DNMT1 [93]. Kaempferol epigenetically reactivates DACT2 transcription to inhibit nuclear β-catenin expression, which in turn inhibits Wnt/β-catenin signaling, which restricts CRC cell proliferation and migration. Additionally, kaempferol-demethylated DACT2 reduced tumor load in AOM/DSS-induced CRC tumorigenesis. Tae et al. found that kaempferol activates autophagy via the inhibition of G9a [94]. According to this study, kaempferol activates IRE1-JNK-CHOP signaling from the cytosol to the nucleus and inhibiting G9a triggers autophagy [95,96,97,98,99,100,101] (Figure 1).

### 7.2. Epigenetic Mechanism of Gallic Acid

Among the emerging cancer treatments is gallic acid (GA) [102,103,104]. Studies have shown it suppresses hepatic cancer cell viability, proliferation, invasion, and angiogenesis [105,106,107,108]. In addition to being an antioxidant, GA is also a pro-oxidant. In H1299 cells, GA reactivates the growth arrest and DNA damage-inducible 45 (GADD45) signaling pathway through demethylation of CCNE2 and CCNB1 [109]. Through a two-week fermentation process, *Aspergillus sojae* is able to efficiently increase the GA content in oolong tea, improving the epigenetic anticancer properties of GA. DNMT1, DNMT3A, and DNMT3B in human cancer cell lines were significantly reduced when the fungus dramatically increased GA up to 45 fold [110]. As a result, fermented oolong tea high in GA appears to be an effective dietary intervention strategy for tobacco-associated cancers because of its potent inhibitory effects on DNMTs. As confirmed by histological analysis, oral administration of GA for 8 weeks decreases tumor size, damages tumor structure, and lowers expression of HDAC1 and PCNA in tumor mass [111]. As an HDAC inhibitor and anti-PCa therapy, GA may inhibit PCa progression by inhibiting HDAC1 and two expressions.

Gallic acid from Rosa rugosa inhibited the majority of histone acetyltransferases (HATs) in another study [112], Including sirtuin, histone deacetylase, and histone methyltransferase. Gallic acid inhibits p300/CBP-dependent HAT activities uncompetitively. Both in vitro and in vivo, GA inhibits p300-induced p65 acetylation. It prevents the nuclear translocation of p65 caused by lipopolysaccharide (LPS). It suppresses LPS-induced nuclear factor-κB activation in A549 lung cancer cells [113] (Figure 2) [110,111,112].

### 7.3. Epigenetic Modulation of Curcumin in Cancer

Curcumin modulates HDAC or HAT activity in a variety of in vitro cancer cell models [114,115,116]. Curcumin inhibits Notch-1 and the pro-inflammatory nuclear transcription factor-kappa B by inhibiting p300-mediated acetylation of its RelA isoform [117]. The researchers showed that curcumin reduces HAT activity in human acute monocytic leukemia THP-1 cells by hypoacetylation p65 at Lys310 and preventing nuclear transcription factor kappaB activity [118]. It has been shown that curcumin induces histone hypoacetylation in brain glioma cells, which results in subsequent apoptotic cell death through PARP- and caspase-3-mediated mechanisms [106]. Recent studies have shown curcumin acts as an epigenetic modulator of TREM-1 gene expression by inhibiting p300 activity in the TREM-1 promoter region, causing hypoacetylation of histones 3 and 4 [118].

During prostate cancer progression in TRAMP mice, Nrf2, the master regulator of cellular antioxidant defenses is epigenetically silenced. Furthermore, curcumin induces hypomethylation of Fanconi anemia (FANCF) promoter that leads to an increase in FANCF protein and gene expression in SiHa cells and a subsequent reduction in ovarian tumor cell proliferation [119]. In MV4-11 leukemia cells, curcumin induces global hypomethylation [120]. Based on molecular docking, curcumin covalently binds to the catalytic thiolate of DNMT1 with an IC50 of 30 nM treatment, causing DNA methylation to be inhibited. More recently, in vitro methylation assays showed that M.SssI, a DNMT1 analog, was inhibited by 50 µM of curcumin in SiHa ovarian cancer cells [108]. This epigenetic event may be related to NF-кB/Sp1 disrupting DNMT1’s promoter. Another study showed that at 5 µM concentration, curcumin also reversed CpG methylation in Neurog1, a cancer methylation marker known to be highly methylated and whose expression is disrupted in human prostate cancer cells [121]. Therefore, curcumin could be a DNMT inhibitor. Using a 2.5 µM dose of curcumin can also restore Nrf2 expression via promoter CpGs demethylation in TRAMP C1 prostate cancer cells [122].

The p300/CREB-binding protein-specific inhibitor curcumin inhibited acetyltransferase activity in HeLa cancer cells when applied at different concentrations [18]. Inhibiting p53 acetylation by histones and p300 leads to apoptosis. In LNCaP, prostate cancer cells treated with curcumin upregulates both phosphorylations at serine and acetylation of p53. The same natural compound induces acetylation of histone H3 and H4 at the same concentration applied for 4 h [123], suggesting that curcumin regulates apoptosis through global regulation of genes related to cell survival and/or apoptosis.

In vivo and ex vivo models of acute myeloid leukemia (AML), curcumin downregulated the expression of DNMT1. DNMT1, p65, and Sp1 become less active for binding to the promoter region of DNMT1 with curcumin. Furthermore, curcumin restored p15INK4b expression by hypomethylating its promoter, resulting in cell cycle arrest in the G1 phase and apoptosis. Curcumin suppressed tumor growth in mice implanted with the MV4-11 cell line of AML [124]. In a TRAMP mouse model of prostate cancer, curcumin inhibited tumor growth by reverting methylation of the Nrf2 promoter. In addition, curcumin inhibited JNK signaling and repressed the H3K4me3 epigenetic mark in LNCaP cells [99]. Curcumin and JQ-1 together suppress prostate cancer development. HT29 colon cancer cells were treated with curcumin and the colony formation and methylation of the DLEC1 promoter were inhibited. Curcumin induced specific changes in DNA methylation of a subset of genes involved in cell viability and proliferation in colorectal cancer cells as opposed to the global hypomethylation induced by 5-aza-CdR [125].

### 7.4. Epigenetic Modulation of Resveratrol in Cancer

Researchers are studying resveratrol (RVT) as a possible treatment for cancer, diabetes, cardiovascular diseases, neurodegenerative diseases, and metabolic diseases [126,127,128,129]. With a focus on epigenetic mechanisms, RVT has biological activities. SIRT1 is downregulated and NF-кB is upregulated in colon cancer (CC), but RVT reverses it [130]. Colon cell lines overexpressing SIRT1 showed antiproliferative effects [131]. By modulating cell cycle-regulating genes, enhancing apoptosis via p53 upregulation, and inhibiting anti-apoptotic genes, RVT inhibited colorectal cancer (CRC) cell proliferation, invasion, and metastasis [132]. In CRC cell lines, NF-кB was down-regulated after RVT. RVT inhibits nuclear translocation of NF-кB when SIRT1 mRNA levels are suppressed, suggesting RVT is SIRT1-dependent. RVT also induced premature senescence in human dermal fibroblasts, which is associated with DNA damage [133]. In vitro and in vivo, RVT has been shown to inhibit tumor growth by affecting the HDAC pathway [134,135,136]. RVT also downregulates MTA1 by acetylating p53 in PCa cells [114]. PTEN, a tumor suppressor, is deleted on chromosome 10 in prostate cancer or inactivated by MTA1/HDAC inhibitors. By abrogating MTA1/HDACs’ negative epigenetic effect, RVT may stimulate PTEN reactivation [137]. Downregulation of MTA1 by RVT triggers apoptosis in prostate cancer cells (PCa) by activating pro-apoptotic genes Bax and p21.

RVT induced apoptosis in Hodgkin lymphoma cells by inhibiting SIRT1 and hyperacetylating p53/FOXO3a [138]. Acyl-p53 and acetyl-FOXO3a levels were increased after RVT treatment, indicating that deacetylase inhibition is a critical step in the apoptosis of lymphoma cells [139]. Hepatoma cell lines treated with RVT showed dose-dependent antiproliferative effects [140]. Another study found the effects of RVT on growth and apoptosis in osteoblastoma cells. Compared with normal osteoblasts, osteosarcoma cells overexpressed the SIRT1 protein after RVT [31].

### 7.5. Epigenetic Modulation of EGCG in Cancer

Worldwide, green tea is widely consumed and believed to prevent cancer. Green tea contains catechins, a type of polyphenol. Most abundant and most biologically active is epigallocatechin-gallate (EGCG). The anticancer properties of EGCG have been extensively studied [141,142,143]. DNA methylation is inhibited by GTPs, which leads to hypomethylation and activation of epigenetically silenced genes. To evaluate the effect of GTPs on DNA methylation, several in vitro experiments have been carried out. A review article summarizes the effect of EGCG on DNA methylation in cell culture. There could be several reasons for the discrepancy between their findings and previous studies, including different methods of analysis, possible gene specificity or cell line specificity of EGCG, or an ineffective treatment method. According to Strassmann et al. [144], in some in vitro cell culture conditions, cellular effects caused by EGCG might be attributed to the oxidative stress caused by it. In an alkaline environment, EGCG undergoes autooxidation, forming homo- and heterodimers of EGCG and EGC, resulting in the formation of H2O2. There is no doubt that auto-oxidation occurs in cell culture and during digestion, but how much it happens depends on the medium [144]. H_2_O_2_ formation, however, was mostly ignored in cell culture experiments. As a result of GTP treatment, silenced genes are re-expressed in in vitro cell culture. DNA methylation will be changed by EGCG concentrations of 20–50 mol/L for 3–6 days. The level is far higher than what mice or humans can physiologically handle [145]. Therefore, GT interventions might not be suitable for therapeutic purposes, but they might affect DNA methylation long term.

Sulforaphane has been shown to enhance EGCG’s DNMT1-inhibitory effect, and at the same time, enhance hTERT inhibition [146]. GT treatment has anticarcinogenic properties in vitro, but EGCG’s limited chemical stability at alkaline pH under normal physiological conditions makes it difficult to translate to the clinic. EGCG and trichostatin A boost ER-α expression in ER-α-negative breast cancer cells [147]. Synthetic analogs have been developed to increase the chemical stability of EGCG, showing stronger anticancer activity and more stability and efficacy [148,149,150]. EGCG inhibits hTERT expression by inducing DNA hypomethylation and promoter deacetylation via DNMTs and histone acetylases, respectively [151]. EGCG and pEGCG inhibited the proliferation of human ER-positive in MCF-7 and ER-negative in MDA-MB-231 breast cancer cells, but not normal cells [152]. In addition, EGCG treatment inhibited the gene and protein expression of DNMT1, DNMT3a, and DNMT3b [153]. Treatment with EGCG led to a significant boost in RECK mRNA expression in oral carcinoma cells (SCC9 and HSC3) [154].

### 7.6. Epigenetic Modulation of Quercetin in Cancer

Flavonoids such as quercetin are found in apples, onions, red wine, and green tea [155,156]. Quercetin causes pro-apoptotic effects in tumor cells without affecting normal ones [157]. DNMT, HDAC, and HMT are all reduced by quercetin in a dose-dependent manner in human cervical cancer cells [158]. Quercetin seems to have an anticancer effect through epigenetic pathways [159,160]. A study used human acute myeloid leukemia cells and HL60 and U937 cell lines. Quercetin treatment eliminated DNMT1 and DNMT3a effects [161]. Quercetin treatment increases apoptosis by DNA demethylation, HDAC inhibition, and H3Ac and H4Ac enrichment in promoter regions of apoptotic genes.

Researchers found quercetin and sodium butyrate downregulated DNMT1, HDAC1, and cyclin D1 [162]. HDAC-NF-кB signaling is inhibited by this combination. Quercetin and curcumin combined treatment resulted in the sensitization of resistant prostate cancer cells to anti-androgen therapy [163]. However, quercetin inhibits HDAC1 and DNMT1 in hamster buccal pouch carcinoma [164], which has a pivotal role in cell survival, and invasiveness, angiogenesis, and cell proliferation. In addition, another researcher found quercetin blocks bound transactivators CREB2, C-Jun, C/EBP*, and NF-кB and the recruitment of the coactivator p300 (Figure 3) [142,143,144,145,146,147,148,149,150,151,152,153,154,155,156,157,158,159,160]. Quercetin inhibited the acetylation of NF-кB by p300 HAT. Another study found that quercetin inhibited colorectal cancer growth by activating p16INK4a induced by promoter demethylation [161]. Quercetin promotes cell death in leukemic HL-60 cells by activating FasL and H3 acetylation. Curcumin and quercetin together restored AR protein levels in androgen-receptor negative prostate cancer cells. DNMT decreased, resulting in global hypomethylation and mitochondrial depolarization, which induced apoptosis.

## 8. Conclusions

Many kinds of cancer are triggered by epigenetic changes. Thus, epigenetic therapy is a valid strategy to stop cancer from spreading. There are a significant number of active molecules in natural compounds, including some that modulate gene expression epigenetically, which deserves more study. DNA aberrations that cause neoplastic transformation can be reversed by many dietary polyphenols. Oncogenes and tumor suppressor genes are epigenetic targets of polyphenol action, which are indirectly controlled by epigenetic enzymes such as DNMTs, HATs, and HDACs. Several polyphenols, including kaempferol, gallic acid, curcumin, resveratrol, EGCG, and quercetin, have been found to downregulate HDAC expression, mostly HDAC 1 and HDAC 3. Polyphenols regulate epigenetic machineries, which may prevent normal cells from turning into tumors. In this review, the majority of studies on polyphenols and epigenetic modifications used cancer cell culture models. The epigenetic modifying effect of these polyphenols on cancer needs further study in vivo.

## Figures and Tables

**Figure 1 ijms-23-11712-f001:**
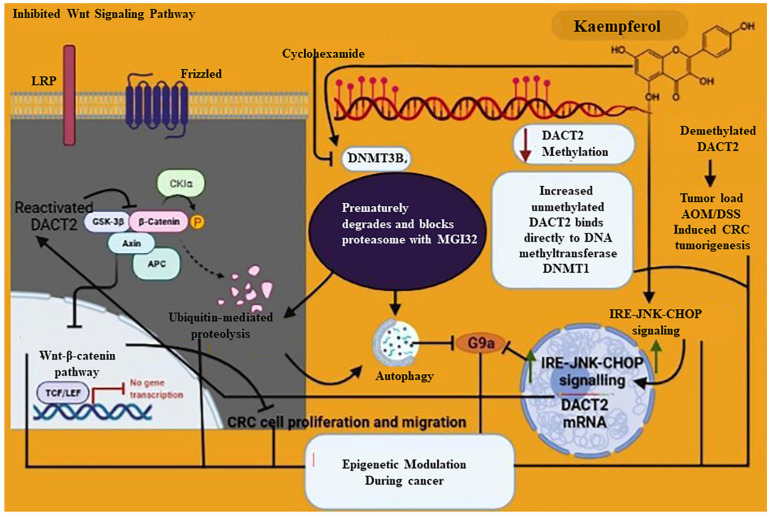
Molecular mechanism represents the role of kaempferol in epigenetic modulation during cancer and how it protects from cancer. Kaempferol protecting against cancer via a different mechanism such as inhibition of DNMT3B which prematurely degrades and blocks proteasome with MGI32 further causes autophagy via G9a inhibition and ubiquitin-mediated proteolysis; on the other hand, reactivated DACT2 inhibits Wnt-β-catenin signaling pathway and leads to impairment of the CRC cell proliferation and migration further causes epigenetic modulation and protects against cancer; increased localization of the IRE-JNK CHOP from cytosolic to the nucleus further inhibits G9a; decreased DACT2 methylation; increased unmethylated DACT2 binds directly to DNA methyltransferase DNMT1 and is involved in epigenetic modulation during cancer.

**Figure 2 ijms-23-11712-f002:**
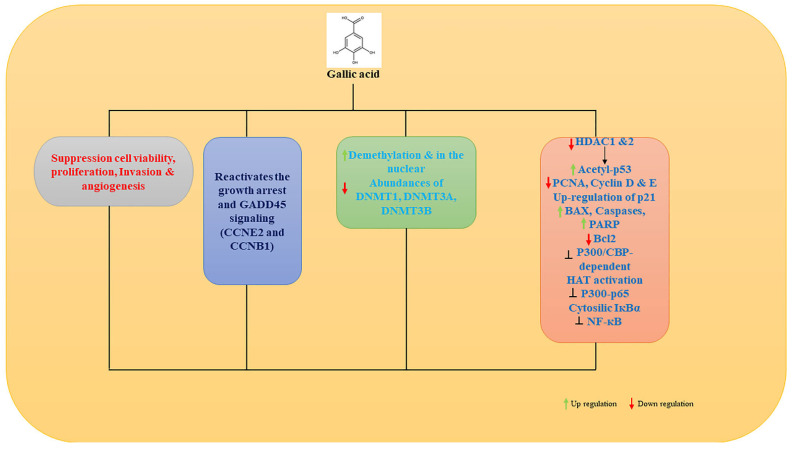
Gallic acid protects from cancer. Gallic acid suppresses cell viability, proliferation, invasion, and angiogenesis; reactivates growth arrest and GADD45; increases demethylation and increases the DNMT1, DNMT3A, and DNMT3B found to be more in the nucleus; decreases HDAC1 and 2; increases acetyl-p53; decreases PCNA; cyclin D and E; up-regulation of p21, decreases BAX, Cas3, and PARP, and increases Bcl2; inhibits P300/CBP-dependent HAT activation; P300-p65 inhibition; cytosolic IKBα; inhibition of NF-kB; and further protects from cancer.

**Figure 3 ijms-23-11712-f003:**
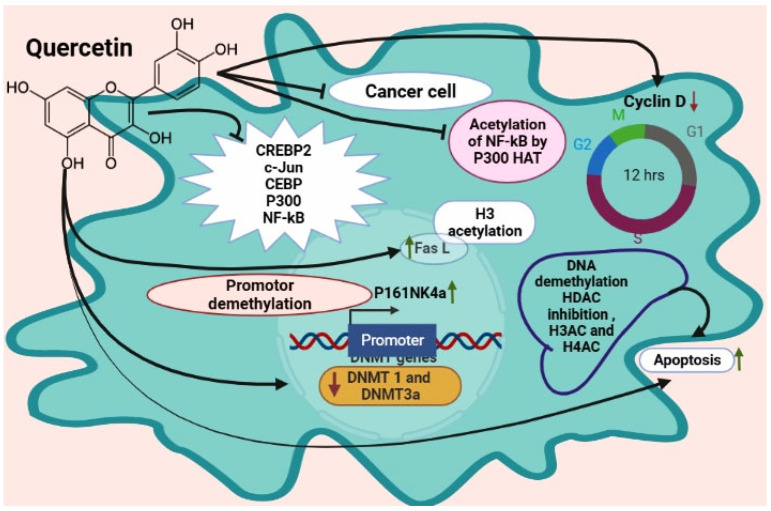
Quercetin protects from cancer via an epigenetic mechanism. Quercetin inhibits NF-kB acetylation by p300 HAT; decreases cyclin D; increases Fas L via H3 acetylation; increases P161NK4a via promoter demethylation; decreases DNMT1 and DNMT3a; DNA methylation HDAC inhibition, H3AC, and H4AC, and increases apoptosis; inhibits the transcriptional CREBP2, c-Jun, CEBP; P300 and NF-kB.

**Table 1 ijms-23-11712-t001:** Polyphenols on DNA methylation and histone modification.

Polyphenols	Molecular Mechanism	Pre Clinical Model	Target Gene
	DNA methylation		
Curcumin	DNMT inhibitor	Leukemia, esophageal	NA
Epicatechin, epicatechin-gallate, epigalocatechin-3-gallate	DNMT inhibitor	Lung, colon cancer cells, esophageal, oral, breast cancers	RAŘ, MGMT, MLH1, CDKN2A, RECK, TERT, RXⱤ, CDX2, GSTP1, W1F1
Quercetin	DNMT inhibitor	Breast, colon, esophageal cancers	CDKN2A
Resveratrol	DNMT inhibitor	Breast, Lungs cancers	NA
	Histone modifications		
Curcumin	HAT and HDAC inhibitor	anti-inflammatory anticancer, antioxidant, antiproliferative	GATA4, EOMES, GZMB, PRF1,H3/H4 deacetylation
Epicatechin, epicatechin-gallate, epigalocatechin-3-gallate	HAT inhibitor	antioxidant, anticancer, anti-inflammatory	NF-kB, IL-6, BMI-1, EZH2, SUZ12, H3K27 trimethylation, H3/H4 acetylation
Quercetin	SIRTI activator HAT inhibitor	anti-migration, anticancer, antiproliferative, antidiabetic, antioxidant	Inflammatory diseases
Resveratrol	SIRTi activator	anti-migration, anticancer, antiproliferative, antidiabetic, antioxidant	TNF-˛, IL-8, RBP

## Data Availability

Not applicable.

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
