# Peer review of "Polyphenols as Potent Epigenetics Agents for Cancer"

_ijms, 2022, doi:10.3390/ijms231911712_

Round 1
Reviewer 1 Report
1. Line 17: “Kaempferol, Curcumin”. The first letter of the chemical composition should be lowercase.
2. Please add a space between the "[]" of the reference and the text, please revise the full text carefully.
3. The secondary titles are all "1.0", please check and revise carefully.
4. Please check references carefully. For example: in lines 43-44, “Diets high in vegetables and fruits have been proven to reduce the risk of cancer[19]. This is because they regulate the expression of oncogenes and tumor suppressor genes.” Both sentences should refer to reference 19. In lines 58-59, “The genes induced by epigenetics were found to be involved in cellular transformation and differentiation, tumor growth, and metastasis[28-30]. Apoptosis, cell adhesion, and cell cycle progression genes are repressed.” The above views should all refer to the references 28-30.
5. lines 125 and 126, caspase-3 is cleaved and activated in the apoptotic cell. “upregulates caspase 3” is a misstatement, “cleaved caspase 3” instead of “caspase 3”.
6. In lines 159 and 160, “Tae et.al. found that kaempferol activates autophagy via the inhibition of G9a.” Missing relevant references.
In lines 160-162, “According to this study, kaempferol activates IRE1-JNK-CHOP signaling from the cytosol to the nucleus and inhibiting G9a triggers autophagy [75,76] (Fig 1).” In the reference 76, “Cinnamaldehyde induces autophagy-mediated cell death through ER stress and epigenetic modification in gastric cancer cells”, its research object is cinnamaldehyde, which has nothing to do with kaempferol. In addition, which references on kaempferol in epigenetic regulation are summarized in Fig 1? Figure 1 cannot be derived only from Ref. 75. Please add the corresponding reference in Figure 1.
7. In lines 180-182, “To improve the epigenetic anti-cancer activities of GA isolated from oolong tea, identified, Aspergillus soda a fungus which can efficiently increase the GA content in oolong tea via a 2-week fermentation process.” “Aspergillus soda” should be italicized. In addition, what is the meaning of this statement? Please rewrite.
8. Please add the corresponding reference in Figure 2 and Figure 3.
indirectly controlled by epigenetic enzymes
To summarize, please first introduce those epigenetic enzymes that phenolic compounds act directly or indirectly, and then further introduce the relevant downstream pathways of cancer prevention. This makes it easier for readers to understand. Please don't just list the content of the literature research, which is inconvenient for readers to understand. If convenient, please add more figures or tables.
Author Response
Reviewer 1
- Line 17: “Kaempferol, Curcumin”. The first letter of the chemical composition should be lowercase.
Thank you for your comments. We have made all changes according to your suggestions in our revised manuscript. Page line 21
- Please add a space between the "[]" of the reference and the text, please revise the full text carefully.
Thank you for your comments. We have made all changes in full text according to your suggestions. Page line 32 to 332
- The secondary titles are all "1.0", please check and revise carefully.
Thank you for your comments. We have made all changes in full text according to your suggestions. Page line 30,52,72,93,108,133,145,175, 202,244,275,309 and 340
- Please check references carefully. For example: in lines 43-44, “Diets high in vegetables and fruits have been proven to reduce the risk of cancer[19]. This is because they regulate the expression of oncogenes and tumor suppressor genes.” Both sentences should refer to reference 19. In lines 58-59, “The genes induced by epigenetics were found to be involved in cellular transformation and differentiation, tumor growth, and metastasis[28-30]. Apoptosis, cell adhesion, and cell cycle progression genes are repressed.” The above views should all refer to the references 28-30.
Thank you for your valuable comments. We have made all changes in full text according to your suggestions. Line number: 47 and 62
- lines 125 and 126, caspase-3 is cleaved and activated in the apoptotic cell. “upregulates caspase 3” is a misstatement, “cleaved caspase 3” instead of “caspase 3”.
Thank you for your valuable comments. We have modified as suggested you.
- In lines 159 and 160, “Tae et.al. found that kaempferol activates autophagy via the inhibition of G9a.” Missing relevant references.
Thank you for your valuable comments. We have modified as suggested you.
Not Tae et al., unfortunately we have changed sur and given name Tae Hoon Kim (Kim TH)
In lines 160-162, “According to this study, kaempferol activates IRE1-JNK-CHOP signaling from the cytosol to the nucleus and inhibiting G9a triggers autophagy [75,76] (Fig 1).” In the reference 76, “Cinnamaldehyde induces autophagy-mediated cell death through ER stress and epigenetic modification in gastric cancer cells”, its research object is cinnamaldehyde, which has nothing to do with kaempferol. In addition, which references on kaempferol in epigenetic regulation are summarized in Fig 1? Figure 1 cannot be derived only from Ref. 75. Please add the corresponding reference in Figure 1.
Thank you for your valuable comments. Sorry for the mistaken we have removed cinnamaldehyde ref. We have added the corresponding references as suggested you. Line number 163.
- In lines 180-182, “To improve the epigenetic anti-cancer activities of GA isolated from oolong tea, identified, Aspergillus soda a fungus which can efficiently increase the GA content in oolong tea via a 2-week fermentation process.” “Aspergillus soda” should be italicized. In addition, what is the meaning of this statement? Please rewrite.
Sorry for typographical error Aspergillus sojae. We have modified in our revised manuscript.
- Please add the corresponding reference in Figure 2 and Figure 3. indirectly controlled by epigenetic enzymes
Thank you for your valuable comments. We have modified as suggested you.
To summarize, please first introduce those epigenetic enzymes that phenolic compounds act directly or indirectly, and then further introduce the relevant downstream pathways of cancer prevention. This makes it easier for readers to understand. Please don't just list the content of the literature research, which is inconvenient for readers to understand. If convenient, please add more figures or tables.
Thank you for your comments. We agree with your most valuable suggestions. We planned to add figures and tables, unfortunately our academic load is very heavy and grant period was very short. If we are not able to publish within the stipulated grant period DSR will never pay the publication fees for the article. We hope you can understand our current situations. Kindly consider and do the needful.
Thank you for all your valuable comments and questions, which allowed us to improve the quality of the manuscript.

Reviewer 2 Report
The paper is well organized and offers a significant overview of the available data (with more than 100 references) on this important topic.
However, I have noticed typos, incorrect chapter numbering, wrong legends under figures 2 and 3, and I also suggest using italics for ''in vivo'' and ''in vitro''.
Author Response
The paper is well organized and offers a significant overview of the available data (with more than 100 references) on this important topic.
We appreciate the time and effort that you and the reviewers have dedicated to providing valuable feedback on our manuscript.
However, I have noticed typos, incorrect chapter numbering, wrong legends under figures 2 and 3, and I also suggest using italics for ''in vivo'' and ''in vitro''.
Thank you very much for your comments. Unfortunately we made mistaken, we have corrected as suggested you in our revised manuscript
Thank you for all your valuable comments and questions, which allowed us to improve the quality of the manuscript.
Reviewer 3 Report
This review article describes our current understanding of dietary polyphenols and how these regulate epigenetic alterations in cancer cells, e.g. DNA methylation and histone modifications, gene expression. Polyphenols may therefore represent an alternative therapeutic means of controlling cancer because of their capacity to modulate epigenetic activity. The manuscript is generally well written and the 3 figures nicely capture the pathways underpinning polyphenol molecular control. There is a substantial amount of information on these compounds making the text particularly dense. The article would be strengthened by incorporating a table listing all the compounds with information regarding their properties and modes of action for quick and simply reference. This should provide a useful resource to the polyphenol community.
Author Response
This review article describes our current understanding of dietary polyphenols and how these regulate epigenetic alterations in cancer cells, e.g. DNA methylation and histone modifications, gene expression. Polyphenols may therefore represent an alternative therapeutic means of controlling cancer because of their capacity to modulate epigenetic activity. The manuscript is generally well written and the 3 figures nicely capture the pathways underpinning polyphenol molecular control. There is a substantial amount of information on these compounds making the text particularly dense. The article would be strengthened by incorporating a table listing all the compounds with information regarding their properties and modes of action for quick and simply reference. This should provide a useful resource to the polyphenol community.
We appreciate the time and effort that you and the reviewers have dedicated to providing valuable feedback on our manuscript
